# The Predictors of Obesity among Urban Girls and Boys Aged 8–10 Years—A Cross-Sectional Study in North-Western Poland

**DOI:** 10.3390/ijerph17186611

**Published:** 2020-09-11

**Authors:** Joanna Ratajczak, Elzbieta Petriczko

**Affiliations:** 1Department of Physical Culture and Health, University of Szczecin, 71-065 Szczecin, Poland; 2Department of Pediatrics, Endocrinology, Diabetology, Metabolic Disorders and Cardiology of Developmental Age, Pomeranian Medical University, 71-242 Szczecin, Poland; elzbieta.petriczko@pum.edu.pl

**Keywords:** epidemiology of childhood obesity, sociodemographic and health factors

## Abstract

Background: Children worldwide are increasingly becoming overweight and obese and developing related health problems, including hypertension, lipid disorders, abnormal glucose tolerance, type 2 diabetes, and secondary psychological disorders. The aim of the study was to determine sociodemographic risk factors that predict an increase in BMI in children at an early school age. Material and method: The study covered 4972 children aged 8–10 years, including boys (N = 2461) and girls (N = 2511). Measurements of basic anthropometric indicators were used, such as body height, body weight, body composition, and physical fitness. The criteria developed by the International Obesity Task Force (IOTF) were adopted. Sociodemographic features were analyzed based on a diagnostic survey. IBM SPSS Statistics v.25 (Mineral Midrange SA, Warsaw, Poland) and IBM SPSS Amos software (Mineral Midrange SA, Warsaw, Poland) were used to perform descriptive statistics, the Kolmogorov-Smirnov test, Pearson′s chi-square test, Student’s *t*-test, and the Mann-Whitney U test. The statistical significance index was assumed to be *p* < 0.05, while *p* < 0.01 was taken as an indicator of a trend which was not completely statistically significant. Results: Both the children and their parents had mainly moderate BMI. A total of 78.7% of children were within the weight norm. Among girls, extreme obesity was two times more frequent than extreme underweight. The examined boys were significantly taller, heavier, and had a higher BMI than girls. There were significant differences between boys and girls in BMI; however, gender alone accounted for less than 1% variance. The influence of parents′ characteristics was much greater, increasing the explained variance to 10%. Body weight of mothers and fathers (*p* < 0.001), mother′s height (*p* < 0.01) and both parents′ level of education (*p* < 0.001) were detected as significant predictors of children’s BMI. Conclusions: The analysis of selected sociodemographic and health factors determining the BMI of the child population indicates the need for preventive action and health promotion both among children and their parents.

## 1. Introduction

Obesity is a chronic systemic disease that significantly shortens human life and is a recognized factor leading to many health disorders [1]. Its growing trend is remarkable: It is predicted that by 2030 the number of obese people living in the U.S. alone will have grown by 65 million and in the UK by 11 million, resulting in an additional 6–8.5 million cases of diabetes, 5.7–7.3 million cases of heart disease and stroke, and 492,000–669,000 additional cancer cases for both these countries [2].

One of the most alarming aspects of this problem is child obesity. It is estimated that there are over 22 million obese children under the age of five [3] and about 10% of the world population under 18 are overweight or obese. In Europe, about 20% of children are overweight and 5% are obese [4,5]. The WHO Childhood Obesity Surveillance Initiative (COSI) research shows that one in three children in the European Union aged six–nine in 2010 was overweight or obese [6]. Significantly, the problem of excessive body weight starts in early childhood. For example, research from the Institute of Mother and Child in Poland indicates that 55% of Polish children aged 13–36 months already have an abnormal body weight [7].

In Poland, the prevalence of obesity ranges from 2.5% to 12% of children and youth, depending on the region. The problem of overweight and obesity affects a total of approximately 12%–14% of children, with considerable regional variation [8]. A study of a representative group of children aged seven–nine years reported overweight and obesity in 15.8% of girls and 15% of boys, with obesity found in 3.7% of girls and 3.6% of boys. Data from the Polish National program for the Prevention and Treatment of Obesity shows a significant proportion of overweight or obese ten-year-old boys (22.4%) and nine-year-old girls (22.4%). These numbers are lower in adolescents (13–15 years), with overweight or obesity found in 14.2%–19.3% of boys and 12.8%–14.2% of girls. In the 16–18 year old age bracket, the percentage of overweight or obese adolescents is lower, ranging from 13.6% to 18% for boys and from 9.1% to 10.9% for girls [9]. The research conducted by the Polish Society of Dietetics on a group of over 14,000 people revealed that slightly more than 76% of junior high school and high school students had the correct body weight, while overweight and obesity were found in about 18% of students [10]. An Percentile charts of height, body mass and body mass index in children and adolescents in Poland–results of the OLAF study on a randomly selected group of 17,573 children and adolescents aged 7–18 years found overweight and obesity in 18% of boys and 14% of girls [11]. An epidemiological study conducted by Kwilosz and Mazur (2016) on a group of 1012 children attending primary schools in the Bieszczady Mountains (average age 9 ± 2 years) presented similar results. Based on OLAF BMI percentile charts, the prevalence of overweight and obesity in the study group was 17.6%; interestingly, excessive body weight was more prevalent among girls (55%) than boys (45%) [12].

Overweight children more frequently suffer from the diseases included in the metabolic syndrome as defined by Reaven [13]. For example, a study on Polish children showed abnormal lipid profile and elevated arterial pressure in more than one in four children with excess body weight. Hyperhomocysteinemia was diagnosed in one third of the examined children, which indicates the possibility of premature development of atherosclerosis and the occurrence of future cardiovascular diseases. The authors emphasize the high rate of metabolic disorders in patients with obesity and that excessive body weight in children should be considered as a health problem, while childhood obesity should be seen as a disease and not as a cosmetic defect. In addition, compared to their healthy peers, obese children more frequently show symptoms of physiological abnormalities, including reduced exercise tolerance, accelerated growth and maturation, as well as steatosis of the liver and gallstones, postural defects, and problems with self-acceptance [14]. A study by Kwieciński et al., (2018) indicates a significant non-linear relationship between BMI and fitness in boys and girls. The results suggest that overweight and obese adolescents generally do not perform well in fitness tests [15]. Childhood obesity increases the risk of many diseases in adult life, e.g., stroke, myocardial infarction, development of other heart diseases, kidney, and locomotor system diseases, sleep apnea, emotional disorders (low self-esteem, depressive states), and some cancers [16,17,18,19]. This shows the significance of early diagnosis of metabolic syndrome in children, application of primary prophylactic measures, and adequate treatment. As atherosclerotic lesions can start in childhood, preventive measures should be implemented very early. The main actors responsible for carrying out preventive interventions are local and national political authorities [20]. In addition, a growing number of researchers indicate the need for interdisciplinary care, i.e., treatment that involves the collaboration of a pediatrician with an endocrinologist, psychologist, psychiatrist, physiotherapist, and nutritionist [21].

The health consequences of childhood obesity are confirmed by a systemic review by Antwi et al. (2012). In addition to obesity-related diseases, the authors draw attention to the enormous economic consequences of obesity for families, health care systems, and the global economy. Direct medical costs of preventive, diagnostic, and therapeutic services related to overweight and associated diseases cover 2%–8% of European health care budgets, a figure which represents 0.6% of their gross domestic products. In the United States, estimates based on 2008 data indicate that overweight and obesity account for USD 147 billion of total medical expenditure, a considerable increase from the $117 billion spent in 2000. Importantly, the indirect costs of overweight and obesity can be much higher, but they are often overlooked. First, the costs of childhood obesity which then continues into adulthood can result in loss of income due to reduced productivity, limited opportunities and reduced activity, illness, absenteeism, and premature death. In addition, there are high costs associated with extensive infrastructure changes that societies need to make to deal with obese people, e.g., reinforced beds, operating tables and wheelchairs, changes to public places, changes in transport safety standards. According to the authors, obesity is reaching pandemic proportions and its consequences will impose a heavy health, financial, and social burden on global society [22].

Searching for the exact causes of overweight and obesity requires broad and systematic analysis of risk factors to identify predictors of overweight in specific environments, assess the trend of change, and to take targeted preventive actions. As the treatment of obesity and overweight in children is based on dietary treatment, change of eating habits and inclusion of physical activity [23,24,25], the quantitative and qualitative differences in potential obesity risk factors among children provides an opportunity to develop targeted health programs and create a more effective nutritional education and special physical activity program for both overweight children and their parents. However, researchers propose many different approaches to risk factors. Risk factors include individual characteristics, characteristics of the social environment, and the effects of their interaction, which are associated with an increased risk of abnormalities, disorders, diseases or premature death. Within these, we can distinguish non-specific factors (e.g., poverty) whose actions may cause many disorders, and specific factors, increasing the probability of certain types of problems or disorders [26]. From the epidemiological point of view, a risk factor is a characteristic or property of an individual or population that appears early in life and is associated with an increased risk of future disease development [27]. Thanks to the statistical approach to risk factors describing them as correlates or predictors of the examined behaviors, we are able to estimate the probability of risky behaviors or diseases [26].

Research on obesity and risk factors in adult obesity also shows the need for an integrative approach to the relationships between factors, the incidence of diseases, and the effect of the environment in which children live. The recommendations for Poland published by the WHO (2017) [28] regarding the prevention of childhood obesity presents a comprehensive approach integrating initiatives in many areas, in line with the postulates of McKinsey′s report that the success of anti-obesity policies requires multiple integrated initiatives. The WHO draws attention to the significance of proper healthcare during pregnancy and breast-feeding and recommends the introduction of taxation on sweetened beverages and the inclusion of obesity prevention into the list of the school nurse’s duties [29].

The growing need to effectively counteract childhood overweight and obesity has informed the local program “Counteracting overweight and obesity among children aged 8 years attending the primary schools in the city of Szczecin”, in short, “The Brave Eight” (*Odważna 8*), financed by the City of Szczecin in the years 2016–2018. The idea of the program was to apply the results of scientific research and to develop knowledge of problems relevant to public health interventions. This means that the extent and nature of the problem should not only be known to specialists in the field, but also to officials directly involved in planning and implementing prevention programs, especially the employees of local governments and the local bodies of the national government. Public health programs are most effective when they concern the specific problems of local communities [30]. Therefore, the City of Szczecin, together with the Pomeranian Medical University and the University of Szczecin, decided to collaborate on the program whose results provide an opportunity to analyze the risk factors of overweight and obesity in the local population of early schoolchildren.

The aim of the study is to present the epidemiology of overweight and obesity in the studied population and to examine the relationship between BMI and birth, child health, and socio-geographic factors.

## 2. Material and Methods

The study involved 4972 children aged 8–10 years, born between 2008 and 2010, including 2461 boys (49.5%) and 2511 girls (50.5%). The respondents were the pupils and parents of 51 public and non-public primary schools in the city of Szczecin in the Zachodniopomorskie Voivodship (NW Poland). The research took place from December 2016 to May 2018. All adult respondents gave their consent to the participation of their child in the study. An appropriate consent was obtained from the Bioethics Committee of the Pomeranian Medical University, Resolution No. KB-0012/150/17 as part of the research project “Counteracting overweight and obesity among children aged 8 years attending primary schools in Szczecin”. The research was carried out as part of the health policy program of the same name financed by the City of Szczecin.

The study included measurements of basic anthropometric indicators such as body height (measured with a stadiometer with an accuracy of +/−0.1 cm), body weight (measured with a scale with an accuracy of +/−100 g), and analysis of body composition (using the electrical bioimpedance method). Physical performance was also measured using the Kasch Pulse Recovery Test step-test, consisting of rhythmic climbing on a platform 30.5 cm high for 3 min enabling the assessment of physical fitness on the basis of heart rate (HR) frequency, and thus the level of aerobic physical activity, the basic indicator of a healthy lifestyle [31]. Based on the measured anthropometric parameters the BMI ((body mass index), kg/m^2^) was calculated. The criteria developed by the International Obesity Task Force (IOTF) were adopted [32].

In the study of sociodemographic features, the diagnostic survey method and the questionnaire of the Polish Society of Health program concerning lifestyle [33] were used. The questionnaire contained 26 questions concerning the child’s health, nutrition, frequency of the child’s physical activity, quality of sleep, fatigue, body weight at birth, the week of pregnancy at birth, and the education, height and weight of the mother and father. The sociodemographic questionnaire is divided into four factors or dimensions for each step in the hierarchical regression model.

The statistical analysis was conducted using the IBM SPSS Statistics v.25 package. The Kolmogorov-Smirnov test of normality was used to determine differences between the assumed BMI distribution and the normal distribution typical for a general population. Pearson’s chi-squared and Mann-Whitney′s U tests were used to compare the examined boys and girls in terms of selected sociodemographic, family, health and lifestyle characteristics. The description of results is complemented by the presentation of median (Mdn) and interquartile range (IQR). However, the main statistical method was a hierarchical regression model used to detect the significance of blocks of potential BMI predictors: the child’s gender, parent’s characteristics, general health indicators and lifestyles. Secondly, the model of regression was used to analyze interactions between the child’s gender and all other BMI predictors. Single data gaps were left unfilled. The value of *p* < 0.05 was taken as an indicator of statistical significance, while the value of *p* < 0.1 was taken as an indicator of a not fully significant statistical trend.

## 3. Results

The exact characteristics of the group in terms of sociodemographic features and basic anthropometric measurements are presented in Table 1. The distributions of all quantitative variables were significantly different from normal: skewed to the right and leptokurtic, which means that in the studied population both children and their parents were dominated by people with a moderate BMI, with a total of 78.7% of children in the weight norm, albeit those with extreme obesity were twice as prevalent than those who were extremely underweight.

As shown in Table 1, the studied boys, compared to girls, were characterized not only by being significantly taller and heavier (*p* < 0.001) but also with having a higher BMI (*p* < 0.01).

A total of 79.6% of boys and 77.9% of girls had a BMI in the norm (between the 5th and 85th percentiles), and 3.7% were underweight (BMI < 5th percentile). 17.6% of pupils were overweight (>85th percentile), including 8.6% obese (>95th percentile). Excessive body weight was more frequent among boys, while extremely low body weight, below the 5th percentile, was observed more often (almost two times more) among girls than in boys (*p* < 0.001). Boys were also characterized by a higher content of muscle tissue and water (*p* < 0.001), while girls were characterized by a higher content of fat (*p* < 0.001). The assessment of the cardiopulmonary capacity [29] of the schoolchildren on the basis of post-workout capacity showed that among all students exactly 65% achieved a result in the norm, and the distributions in both genders were statistically equal.

Table 2 contains information from an interview with the parents of the examined children, concerning their general health and eating habits. Sex-related differences were observed in many of the examined factors. More than 97% of parents assessed their children’s health as good or very good, although boys had better cardiopulmonary capacity (*p* < 0.001) and general health (*p* < 0.01). No chronic diseases were declared by 67.1% of boys and 72.4% of girls (*p* < 0.001). The most frequent ailments were allergies (16.3%), which were 2.6% more frequent in boys (*p* < 0.05); boys also had asthma two times more often than girls (*p* < 0.001). None of the students declared diabetes and intestinal diseases. Moreover, the boys were more often under the care of an allergologist, pulmonologist (*p* < 0.001), or neurologist (*p* < 0.05).

The nutritional interview shows that over 45% of the students consumed sweets several times a week, and only 0.3% of girls and 0.6% of boys did not eat sweets at all (*p* > 0.05). Boys consumed more fast food (*p* < 0.001), although they were more physically active (*p* < 0.01) and less frequently complained about sleep problems (*p* < 0.001) and tiredness (*p* < 0.01). More than every fifth girl and almost every fourth boy watched TV for more than 3 h a day on average (*p* < 0.001). It is worth noting that 3.5% of parents did not know if their children ate fast-food at all, and 0.4% did not know whether their children ate sweets or not.

In order to determine whether the examined sociodemographic, family, health and nutritional factors were responsible for the increase in BMI in children, a hierarchical regression analysis was conducted, which revealed the existence of a dozen or so predictors of obesity in early school age. Table 3 presents the results of the hierarchical model for boys and girls. All “don’t know” answers were removed from potential predictors and the values were standardized.

There existed significant differences between boys and girls in BMI; however, gender alone accounted for less than 1% variance. The influence of parents’ characteristics was much greater, increasing the explained variance to 10% (F (6) = 97.239; *p* < 0.001). Body weight of mothers and fathers (*p* < 0.001), mother’s height (*p* < 0.01) and both parents’ level of education (*p* < 0.001) were detected as significant predictors of children’s BMI. Every 1 kg increase in mother’s body weight resulted in a 0.17 increase in BMI of their children, and an increase in fathers’ body weight by 1 kg resulted in a BMI increase of 0.13 points. The third block of potential predictors, i.e., child’s general condition, increased the percentage of total explained variance to nearly 13% (F (4) = 30.629; *p* < 0.001]. Child’s BMI was significantly affected by their health status, shorter duration of pregnancy and higher birth weight. The last block, lifestyle characteristics, increased the total explained variance to almost 15% (F (9) = 11.881; *p* < 0.001). Children’s BMI increased accordingly with higher frequency of the consumption of sweets and duration of watching TV, as well as lower frequency of physical activity, greater consumption of breakfasts and water intake, including sweetened water. The inclusion of lifestyle factors did not change the previously detected prediction strength of gender, parents’ characteristics and factors related to the child’s general condition.

An additional model of regression concerning the interaction between the child’s gender and other predictors showed that only the influence of TV-watching duration on BMI was significantly moderated by gender; watching TV seems to be important only for boys (beta = 0.042; *p* = 0.002, while duration of physical activity is more significant for girls (beta = −0.030; *p* = 0.028). Some statistically non-significant differences in BMI prediction were related to fast-food consumption (beta = 0.023; *p* = 0.090), sleeping problems (beta = 0.025; *p* = 0.076) and father’s body weight (beta = 0.024; *p* = 0.076), which were slightly stronger for boys than for girls.

## 4. Discussion

Our regional survey of 4972 children aged 8–9 indicated that 17.6% of the students were overweight, including 8.6% who were obese. Boys more frequently had excessive body weight, and girls two times more frequently showed extremely low weight (below the 5th percentile). This confirms previous scientific research conducted on different populations of children. At the same time, it indicates that in spite of the prophylactic measures carried out no change has occurred in recent years in the proportion of children with excessive body weight.

Our study aimed to determine whether selected sociodemographic, family, health and nutritional factors may be used as predictors of elevated BMI in children. There were significant differences between boys and girls in BMI; however, gender alone accounted for less than 1% variance. The influence of parents’ characteristics was much greater, increasing the explained variance to 10%. Body weight of mothers and fathers (*p* < 0.001), mother’s height (*p* < 0.01) and both parents’ level of education (*p* < 0.001) were detected as significant predictors of children’s BMI. Every 1 kg increase in mother’s body weight resulted in a 0.17 increase in BMI of their children, and an increase in fathers’ body weight by 1 kg resulted in a BMI increase of 0.13 points. The third block of potential predictors—child’s general condition—increased the percentage of total explained variance to nearly 13%. Child’s BMI was significantly affected by health status, shorter duration of pregnancy, and higher birth weight. The last block, i.e., lifestyle characteristics, increased the total explained variance to almost 15%. Children’s BMI increased with the higher frequency of the consumption of sweets and duration of watching TV, as well as lower frequency of physical activity, greater consumption of breakfasts and water intake (including sweetened water). The inclusion of lifestyle factors did not change previously detected prediction strength of gender, parents’ characteristics and factors related to the child’s general condition.

Boys and girls showed certain differences in their health status reported by their parents. More than 97% of parents assessed their children’s health as good or very good, although boys had a better cardiopulmonary function and general health. No chronic diseases were declared by 67.1% of boys, compared to 72.4% of girls. Among reported diseases, the most common included allergies, which were more frequent in boys (boys also suffered from asthma two times more often). None of the pupils had diabetes or intestinal diseases. In addition, boys were more often under the care of an allergologist, pulmonologist, or neurologist than girls. Our family food interview showed that more than 45% of the schoolchildren consumed sweets several times a week and only 0.3% of girls and 0.6% did not eat sweets at all. Boys consumed more fast food, although they were significantly more physically active and less often had problems with sleep and chronic fatigue. More than every fifth girl and almost every fourth boy watched TV for more than 3 h a day on average.

Our data generally confirm the results of other authors on lifestyle which indicate the significance of family eating habits for the BMI of children. The results of the study by Kurylak et al., (2009) showed that most of the parents surveyed admitted that they did not attach particular importance to a healthy diet and that as a family they would rather lead a sedentary lifestyle. In one in four families, at least one person could be characterized by the considerable intake of nicotine, alcohol or other intoxicants. In the surveyed families, relatively little importance is attached to healthy behaviors (such as following a healthy diet, leading an active lifestyle and preferring active rest). As far as nutrition is concerned, it is predominantly based on one’s own experience and habits passed on from one generation to the next; specialist knowledge is rarely applied here. When the children were asked under what circumstances they tend to eat more or more often, they cited boredom, stress, and the communal experience of eating chips, fries, and sweets with their friends. Parents, when asked about how they plan their children’s diets, most often cited their own experience and family tradition. The use of academic knowledge, books and specialist articles, or consultations with a physician/dietician, was declared only by a minority [34].

The influence of nutrition, TV watching and physical activity on BMI, which we found to affect our study population’s BMI, is confirmed by Tabak et al., (2012) who analyzed the relationship between the frequency of family meals and body weight of 13-year-olds and selected determinants in a group of 605 children. Most young people (80%–90%) ate each of their main meals in the company of their parents at least once a week, 21% ate breakfast with their parents every day, 41% had lunch, and 45% had supper. The frequency of family meals and the number of hours spent watching TV or watching a computer correlated negatively with the BMI of girls, and positively with physical activity, regular meals and the consumption of vegetables by young people of both sexes. The lowest average BMI values were found in the group of adolescents eating family meals frequently, the highest in the group of young people who rarely ate family meals (over 20% of young people in this group were overweight), but the differences were statistically significant only for girls (*p* = 0.025). According to the authors, family meals treated as a predictor of a healthy lifestyle may indirectly protect young people from overweight and obesity. Therefore, family meals should be promoted as an important method of preventing obesity, especially among teenage girls [35].

Scientific analyses have studied the socioeconomic situation of the family and its impact on becoming overweight. Sundblom et al. (2008) studied time trends of overweight, obesity and underweight in ten-year-old children, highlighting gender and areas with different socio-economic status as contributing factors. The results indicated higher obesity and overweight in socioeconomically disadvantaged areas, for both genders [36]. Studies on Portuguese children confirm our results that the weight of parents is an important predictor of overweight and obesity in children. The authors showed that paternal obesity was an important predictor of obesity in children. Obese fathers usually served their children meals that were more calorific [37]. A study by Ohly et al., (2013) conducted in Cornwall (rural southwestern England) and Islington (London borough) showed that less educated parents required assistance and support regarding healthy eating habits [38].There are also studies that confirm the risk of developing obesity in the pediatric population due to high birth weight (over 4000 g), (Dubois, 2006), and intensive weight gain between birth and five–six months of life [39,40].

The analysis of overweight risk factors in the pediatric population was conducted by Wieniawski and Werner (2012). According to the authors, risk factors and states of high risk of overweight and obesity in children include a strong positive family history of premature cardiovascular diseases or incidents, overweight or obesity and other components of metabolic syndrome, hypertension, male sex, coexistence of low HDL (high-density lipoprotein) cholesterol, high triglyceride and LDL (low-density lipoprotein) concentrations, conditions increasing the risk of atherosclerosis (e.g., diabetes), HIV infection, systemic lupus erythematosus, post-transplant and post-neoplastic condition, nicotinism or passive exposure to tobacco smoke and, the presence of new risk factors such as elevated levels of lipoprotein (a), homocysteine, or C-reactive protein. In their summary, the authors emphasize the importance in the prevention of childhood obesity of parents who are at an increased risk of cardiovascular diseases (parents with obesity and/or hypertension and/or diabetes). They advise the systematic monitoring of children’s body weight and the education of children and parents about the need to change their lifestyle and diet and increase physical activity [21].

Our results indicate the need for further in-depth research on the epidemiology of overweight and obesity and the relationship between BMI and sociodemographic, birth and child health factors in the context of current health problems. The understanding of the mechanisms behind the obesity pandemic gives an opportunity to plan adequate health education policies and create highly specialized health programs with a solution adequate to a specific community. This is consistent with the results of Jang’s review (2015), which shows that interventions targeting parents with information about healthy habits for children can be effective in reducing overweight and obesity in children in the short and long term [41,42].

### Limitations

The strengths of our research include the sample size and information on the lifestyle of children and parents. The limitations consist in the lack of measurement of the relationship between parental perception and the actual health and fitness of their children. We plan to address this issue in our future research.

## 5. Conclusions

In the study, 78.7% of children were within weight norms, but more than two times more were characterized by extreme obesity than by extreme underweight. Overweight was more frequent among boys, while among girls the proportion of those with extremely low weight was two times more frequent than in boys.

The study revealed the existence of obesity predictors in early school age, showing certain quantitative and qualitative differences between boys and girls. The model for boys (10 predictors) differed in some respects from the model for girls (11 predictors), with some predictors common for both groups. For both groups, the strongest predictors were the parents’ body weight, frequency of physical activity, general health, birth weight, time spent watching TV, eating sweets, and a lower height in the mother, although the strength of these predictors differed.

The BMI of boys was affected by the height of the mother, an earlier week of birth, more frequent fast food consumption and less frequent drinking of water. The girls’ BMI was also affected by the frequency of drinking water and sweetened water. An important predictor was also the education of parents; a higher BMI of boys correlated with a lower level of education of fathers, while the girls’ BMI showed the same type of correlation with the lower education of mothers.

Importantly, it was shown that both the weight of the mother and the father significantly correlated with many predictors of the behavior and nutrition of children.

Our results indicate the primary significance of prophylactic measures among parents in order to achieve the correct BMI of children. This means that official programs aiming at combating overweight and obesity should focus on actions targeting entire families, and not only children, whose eating habits and lifestyle depend largely on the choices made by their parents.

## Figures and Tables

**Table 1 ijerph-17-06611-t001:** Characteristics of the examined children including gender differences, N (%) or M (SD).

Variables	Whole Group, N = 4972	Boys, N = 2461	Girls, N = 2511	*p*-Value
Age, years	9.3	9.3	9.2	<0.001
Body height, cm	131.00 (10.00)	132.00 (10.00)	130.00 (9.00)	<0.001
Body weight, kg	28.40 (8.00)	28.70 (9.00)	27.90 (8.00)	<0.001
BMIcentile, kg/m^2^	52.00 (98.00)	50.00 (51.00)	55.00 (54.00)	0.028
BMI intervals; N (%)				
below 5 percentile	185 (3.7)	62 (2.5)	123 (4.9)	<0.001
below 15 percentile	550 (11.1)	252 (10.2)	298 (11.9)	
below 50 percentile	1850 (37.2)	966 (39.3)	884 (35.2)	
below 85 percentile	1513 (30.4)	740 (30.1)	773 (30.8)	
below 95 percentile	446 (9.0)	209 (8.5)	237 (9.4)	
above 95 percentile	428 (8.6)	232 (9.4)	196 (7.8)	
Post-workout	122.40 (14.36)	118.88 (14.05)	125.84 (13.81)	<0.001
Cardiopulmonary capacity N (%)				
Very poor	353 (7.1)	218 (8.9)	135 (5.4)	<0.001
Poor	1262 (25.4)	593 (24.1)	669 (26.6)	
Sufficient	1326 (26.7)	628 (25.5)	698 (27.8)	
Good	963 (19.4)	537 (21.8)	426 (17.0)	
Very good	608 (12.2)	293 (11.9)	315 (12.5)	
Excellent	106 (2.1)	50 (2.0)	56 (2.2)	
Vaccination N (%)		2310 (97.2)	2345 (97.5)	0.591
Mother′s body height; cm	166.00 (7.00)	166.00 (8.00)	166.00 (7.00)	0.688
Father′s body height; cm	180.00 (10.00)	180.00 (10.00)	180.00 (10.00)	0.313
Mother′s body weight kg	66.00 (14.00)	64.00 (14.00)	64.00 (14.00)	0.273
Father′s body weight; kg	87.00 (15.00)	87.00 (16.00)	87.00 (15.00)	0.758
Duration of pregnancy; the length of the pregnancy months	39.00 (2.00)	39.00 (2.00)	39.00 (2.00)	0.054
Birth weight g	3359.00 (580.00)	3400.00 (600.00)	3359.00 (550.00)	<0.001
Mother′s education; N (%)				0.265
Primary	141 (3.0%)	65 (2.8%)	76 (3.2%)	
Vocational	405 (8.6%)	203 (8.7%)	202 (8.5%)	
High-school	1191 (25.2%)	565 (24.1%)	626 (26.3%)	
University	2989 (63.2%)	1510 (64.4%)	1479 (62.1%)	
Father’s education; N (%)				
Primary	153 (3.4)	72 (3.2)	81 (3.5)	0.448
Vocational	783 (17.2)	392 (17.4)	391 (17.1)	
High-school	1446 (31.8)	696 (30.8)	750 (32.7)	
University	2168 (47.6)	1098 (48.6)	1070 (46.7)	

Source: own study.

**Table 2 ijerph-17-06611-t002:** Characteristics of health and nutrition of the examined boys and girls N (%) or M (SD).

Variables	Whole Group, N = 4972	Boys, N = 2461	Girls, N = 2511	*p*-Value
Cardiopulmonary capacity; N (%)				
No measurement	354 (7.1)	142 (5.8)	212 (8.4)	<0.001
Excellent	106 (2.1)	50 (2.0)	56 (2.2)	
Very good	608 (12.2)	293 (11.9)	315 (12.5)	
Good	963 (19.4)	537 (21.8)	426 (17.0)	
Sufficient	1326 (26.7)	628 (25.5)	698 (27.8)	
Poor	1262 (25.4)	593 (24.1)	593 (24.1)	
Very poor	353 (7.1)	218 (8.9)	135 (5.4)	
General health; N (%)				
Very good	2632 (55.8)	1267 (54.2)	1365 (57.4)	0.008
Good	1949 (41.3)	984 (42.1)	965 (40.6)	
Moderate	111 (2.4)	70 (3.0)	41 (1.7)	
Bad	3 (0.1)	2 (0.1)	1 (<0.1)	
Difficult to say	21 (0.4)	14 (0.6)	7 (0.3)	
Chronic diseases; N (%)				
No	3482 (70.0)	1665 (67.1)	1817 (72.4)	<0.001
Diabetes	0 (0.0)	0 (0.0)	0 (0.0)	-
Asthma	192 (3.9)	126 (5.1)	66 (2.6)	<0.001
Thyroid	77 (1.5)	36 (1.5)	41 (1.6)	0.648
Intestines	0 (0.0)	0 (0.0)	0 (0.0)	-
Allergies	808 (16.3)	432 (17.6)	376 (15.0)	0.014
Respiratory	54 (1.1)	27 (1.1)	27 (1.1)	1.000
Cardiovascular	53 (1.1)	25 (1.0)	28 (1.1)	0.783
Neurological	48 (1.0)	31 (1.3)	18 (0.7)	0.062
Gastric	64 (1.3)	29 (1.2)	35 (1.4)	0.531
Endocrinological	31 (0.6)	17 (0.7)	14 (0.6)	0.592
Other	125 (2.5)	64 (2.6)	61 (2.4)	0.718
Permanent medical care of a physician: N (%)				
Allergologist	690 (13.9)	389 (15.8)	301 (12.0)	<0.001
Pulmonologist	221 (4.4)	136 (5.5)	85 (3.4)	<0.001
Endocrinologist	143 (2.9)	72 (2.9)	71 (2.8)	0.865
Cardiologist	100 (2.0)	54 (2.2)	46 (1.8)	0.366
Nephrologist	45 (0.9)	23 (0.9)	22 (0.9)	0.882
Gastrologist	89 (1.8)	44 (1.8)	45 (1.8)	1.000
Neurologist	70 (1.4)	44 (1.8)	26 (1.0)	0.030
Other	461 (9.3)	241 (9.8)	220 (8.8)	0.222
Consumption of sweets; N (%)				
Min. 5 times/week.	627 (13.1)	302 (12.7)	325 (13.5)	0.300
Max. 1 time/day	1328 (27.8)	677 (28.6)	651 (27.0)	
Several times/week	2181 (45.6)	1055 (44.5)	1126 (46.7)	
Max. 1 time/week	607 (12.7)	312 (13.2)	295 (12.2)	
Not at all	21 (0.4)	14 (0.6)	7 (0.3)	
I don’t know	18 (0.4)	9 (0.4)	9 (0.4)	
Breakfast; N (%)				
Min. 5 times/week	3712 (78.9)	1853 (79.7)	1859 (78.2)	0.431
3–5 times/week	465 (9.9)	230 (9.9)	235 (9.9)	
1–3 times/week	442 (9.4)	204 (8.8)	238 (10.0)	
None	77 (1.6)	34 (1.5)	43 (1.8)	
I don’t know	6 (0.1)	4 (0.2)	2 (0.1)	
TV–duration				
>3 h daily	1086 (22.9)	568 (24.3)	518 (21.6)	<0.001
>1–3 h daily	2575 (54.4)	1296 (55.4)	1279 (53.3)	
<1 h daily	1076 (22.7)	474 (20.3)	602 (25.1)	
Fast-food; N (%)				
Min. 5 times/week	25 (0.6)	13 (0.6)	12 (0.5)	<0.001
3–5 times/week	43 (1.0)	26 (1.2)	17 (0.8)	
1–3 times/week	1888 (43.0)	999 (46.2)	889 (39.8)	
None	2282 (51.9)	1048 (48.5)	1234 (55.3)	
I don’t know	155 (3.5)	76 (3.5)	79 (3.5)	
Water; N (%)				
Min. 5 times/week	3226 (68.8)	1611 (69.7)	1615 (67.9)	0.176
3–5 times/week	617 (13.2)	284 (12.3)	333 (14.0)	
1–3 times/week	546 (11.6)	257 (11.1)	289 (12.2)	
None	278 (5.9)	146 (6.3)	132 (5.6)	
I don’t know	20 (0.4)	12 (0.5)	8 (0.3)	
Sweetened water; N (%)				
Min. 5 times/week	701 (15.0)	360 (15.5)	341 (14.5)	0.303
3–5 times/week	585 (12.5)	283 (12.2)	302 (12.8)	
1–3 times/week	1674 (35.8)	847 (36.4)	827 (35.2)	
None	1688 (36.1)	818 (35.1)	870 (37.0)	
I don’t know	32 (0.7)	20 (0.9)	12 (0.5)	
Physical activity; N (%)				
Min. 5 h/week	2001 (42.0)	1049 (44.5)	952 (39.6)	0.001
3–5 h/week	1552 (32.6)	768 (32.6)	784 (32.6)	
1–3 h/week	1078 (22.6)	484 (20.5)	594 (24.7)	
<1 h/week	131 (2.8)	58 (2.5)	73 (3.0)	
Sleeping problems; N (%)				
Never	1954 (40.9)	1038 (43.9)	916 (37.9)	<0.001
Very seldom	2099 (43.9)	1001 (42.3)	1098 (45.4)	
1 time/month	185 (3.9)	92 (3.9)	93 (3.8)	
1–2 times/week	305 (6.4)	138 (5.8)	167 (6.9)	
3–4 times/week	163 (3.4)	66 (2.8)	97 (4.0)	
>5 times/week	77 (1.6)	32 (1.4)	45 (1.9)	
Fatigue; N (%)				
Never	885 (18.5)	463 (19.6)	422 (17.5)	0.001
Very seldom	2468 (51.7)	1244 (52.7)	1224 (50.6)	
1 time/month	421 (8.8)	215 (9.1)	206 (8.5)	
1–2 times/week	740 (15.5)	316 (13.4)	424 (17.5)	
3–4 times/week	227 (4.8)	102 (4.3)	125 (5.2)	
>5 times/week	36 (0.8)	19 (0.8)	17 (0.7)	

**Table 3 ijerph-17-06611-t003:** Hierarchical regression analysis of children’s BMI.

Variables		Step 1	Step 2	Step 3	Step 4
Basic	Gender	−0.038 **	−0.044 **	−0.028 *	−0.027 *
Parent’s characteristics	Mother’s body weight		0.205 ***	0.188 ***	0.181 ***
	Mother’s height		−0.044 **	−0.053 ***	−0.047 **
	Father’s body weight		0.163 ***	0,157 ***	0.148 ***
	Father’s height		−0.023	−0.020	−0.014
	Mother’s education		−0.078 ***	−0.066 ***	−0.051 **
	Father’s education		−0.088 ***	−0.081 ***	−0.068 ***
Child’s general condition	Health status			0.129 ***	0.119 ***
	Cardiopulmonary capacity			0.023	0.016
	Duration of pregnancy			−0.030 *	−0.033 *
	Birth weight			0.087 ***	0.086 ***
Lifestyle characteristics	Consumption of sweets				0.068 ***
	Breakfast				0.032 *
	watching TV-duration				−0.075 ***
	Fast-food				0.011
	Sweetened water				−0.043 **
	Water				−0.050 **
	Physical activity				0.089 ***
	Sleeping problems				−0.009
	Strength				-0.026
Model characteristics	R^2^	0.001	0.106	0.128	0.146
	Adjusted R^2^	0.001	0.105	0.126	0.143
	F	7.014 **	84.466 ***	66.172 ***	42.160 ***
	df	1	7	11	20

* *p* < 0.05, ** *p* < 0.01, *** *p* < 0.001; R2: goodness of model’s fit, F: ANOVA test, df: degrees of freedom.

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
