# Peer review of "The Predictors of Obesity among Urban Girls and Boys Aged 8–10 Years—A Cross-Sectional Study in North-Western Poland"

_ijerph, 2020, doi:10.3390/ijerph17186611_

Round 1
Reviewer 1 Report
This study, conducted in a survey of 8-to-9-year-old children from Poland, aims to determine sociodemographic predictors of weight status in this population. A significant strength of the study is the sample size, as it consists of approximately 5000 male and female participants. Also, the survey provides data on lifestyle habits, which are also included in the analysis. This sort of study is always of interest as the conclusions may serve to inform policymaking and improve the focus of preventive interventions. However, in its current form, I'm not sure the study is ready for publication. Hence, I have several recommendations that were thought to add to the paper and help the authors to work on an improved version of their manuscript.
General comments
The manuscript is mostly well written, but it may greatly benefit from a thorough language editing by someone who is a native speaker.
Specific comments
Methods
- Because the recommended standard for assessment of children's growth is the 2007 WHO reference, I suggest including weight status evaluation using this reference, along with the IOTF definition, at least for descriptive purposes.
- May the authors provide a copy of the questionnaire of the Polish Society of Health, perhaps as supplementary material ar as an Annex? There are lots of variables here for which we do not know the actual definition.
- The Mann-Whitney test compares median values; it does not compare mean values. Please correct this in the methods section.
- If you have variables that lack a normal distribution, you should also use the Spearman correlation coefficient.
Results
- In order to compare boys' and girls' height, weight and BMI, you should use standardized values instead of raw values. One way to so is by estimating z scores using the 2007 WHO growth standard and tables (https://www.who.int/childgrowth/en/). Alternatively, you can compare percentiles. However, you cannot do this sex-comparison using raw values; it's a mistake. Whatever you choose, keep in mind that this might completely change your results.
- Since you have parental height and weight, it should be much better to estimate and report parental BMI instead of height and weight. After all, it is the parental weight status that determines child weight status.
- Children's health status, as mentioned on page 6, paragraph 1, is actually the parental perception of health status. Although this is out of the scope of the study, it would be interesting to measure the relation between perception and the children's health and fitness status. Maybe for future work.
- The finding that 3.5% of parents do not know whether their 8-years-old children eat fast-food items is something that deserves further comments in the discussion section. First, because we are talking here of children, preadolescents children. Second, because 3.5%, although not a vast quantity, it denotes something more than an accident or a random event.
- I recommend defining cardiopulmonary fitness categories, maybe as a footnote under the table.
- Table 3: For the categorical predictors, it would be interesting to estimate the differences between categories.
- Figure 1: I'm a bit confused with this diagram. According to the methods section, you ran linear regressions. Nevertheless, this diagram denotes a SEM model. Linear regressions assume independence of predictors, but here some predictors are clearly related to others. Please, clarify which method is being used to test the relationships of interest in the study.
Discussion
- The discussion section will probably change once the authors rearrange their analysis.
- Because this study focuses on the determinants of overweight/obesity in children, I would have expected some recommendations for policy and practice.
- Also, the study has several limitations that should be reckoned by the authors, along with its strengths.
Author Response
Dear Editor,
Thank you very much for proposed corrections. We have substantially revised the manuscript according to your suggestions.
Below are the detailed responses to the Reviewer's comments.
The manuscript is mostly well written, but it may greatly benefit from a thorough language editing by someone who is a native speaker.
Response: A native speaker has corrected the manuscript. Please see the attached certificate issued by the native speaker.
Because the recommended standard for the assessment of children's growth is the 2007 WHO reference, I suggest including weight status evaluation using this reference, along with the IOTF definition, at least for descriptive purposes.
In order to compare boys' and girls' height, weight and BMI, you should use standardized values instead of raw values. One way to so is by estimating z scores using the 2007 WHO growth standard and tables (https://www.who.int/childgrowth/en/). Alternatively, you can compare percentiles. However, you cannot do this sex-comparison using raw values; it's a mistake. Whatever you choose, keep in mind that this might completely change your results.
Response:. WHO Child Growth Standards is an excellent tool for assessing child's development. In our study, we used the results of the OLAF study concerning the body weight, height and BMI of children and adolescents in Poland. The OLAF research project (PL0080) was implemented thanks to the support provided by Iceland, Liechtenstein and Norway through funding from the European Economic Area Financial Mechanism and the Norwegian Financial Mechanism and the Polish Ministry of Science and Higher Education.
The aim of the OLAF study was to update the reference ranges of growth in the form of percentile and Z-score charts of height, body weight and BMI for schoolchildren and adolescents. The study was based on the anthropometric data of 17 281 children and adolescents aged 6-19 years, participants of the project "Development of blood pressure standards for the population of children and adolescents in Poland" - PL0080 OLAF. Data were analyzed using SAS 9.2 statistical package. The percentile and Z-score charts of anthropometric parameters were developed using the LMS method and the LMS Chart Maker Pro 2.42 package. The study gave the results that are up-to-date, representative for the school age population of children and adolescents, percentile and Z-score charts of body height, body weight and BMI, which enable early detection of developmental and nutritional disorders.
This research tool has been used in other papers, e.g: Szczyrska, J; Jankowska, A.; Brzeziński, M.; Jankowski, M.; Metelska, P.; Szlagatys-Sidorkiewicz, A. Prevalence of Overweight and Obesity in 6-7-Year-Old Children - A Result of 9-Year Analysis of Big City Population in Poland. Int. J. Environ. Res. Public Health 2020, 17, 3480-3489.
The Mann-Whitney test compares median values; it does not compare mean values. Please correct this in the methods section.
Response: The Mann-Whitney U test we used was obviously performed based on median. Means (M) and standard deviations (SD) are given additionally to facilitate the reading of data.
If you have variables that lack a normal distribution, you should also use the Spearman correlation coefficient.
Response: In order to address the slightly different comments of all reviewers, it was decided to change the main calculation method to a hierarchical regression model. Therefore, the correlation analyses have been completely removed from the Results section.
Since you have parental height and weight, it should be much better to estimate and report parental BMI instead of height and weight. After all, it is the parental weight status that determines child weight status.
Response: Response: We do plan to include parents' BMI and additional results regarding children in our future research. Thank you for the comment and suggestions for our further research.
Children's health status, as mentioned on page 6, paragraph 1, is actually the parental perception of health status. Although this is out of the scope of the study, it would be interesting to measure the relation between perception and the children's health and fitness status. Maybe for future work.
Response: Response: We do agree that it would be very interesting to investigate a relationship between the parental perception of the child's health and the actual state of the child's health and fitness. Thank you very much for this indication. We are currently conducting a partial assessment of parental perception. We are still in the process of researching the children's population in our region. The results of this research will be the subject of another study.
The finding that 3.5% of parents do not know whether their 8-years-old children eat fast-food items is something that deserves further comments in the discussion section. First, because we are talking here of children, preadolescents children. Second, because 3.5%, although not a vast quantity, it denotes something more than an accident or a random event.
Response: Currently, together with a team of nutritionists, we are conducting additional research on the nutrition in the overweight children in our region. The results of the research will be published in subsequent works. In the discussion we referred only to the issue of fast food consumption as a health problem in children's populations. This is an extremely interesting issue for our team.
I recommend defining cardiopulmonary fitness categories, maybe as a footnote under the table.
Response: Cardiorespiratory fitness is one measure of body functions, and its assessment should play an important role in the activities associated with the promotion of physical activity as an important component of a healthy lifestyle. This study aimed to develop a reference system of the mean post-exercise heart rate (HRmean post-ex) after a 3-min step test for use in screening the cardiorespiratory fitness of 6- to 12-year-old children. The study included 14,501 children ages 6–12 years from primary schools in Gdansk. The participants were subjected to the 3-min Kasch Pulse Recovery Test (KPR Test). The reference range for the classification of cardiorespiratory fitness was developed on the basis of the age-specific percentile distribution of HRmean post-ex in 6- to 9- and 10- to 12-year-old children. This study showed that the 3-min KPR Test is easy to perform and well tolerated by school-age children. As such, it can constitute a useful tool for health promoters and educators. The presented age- and gender-specific reference range of HRmean post-ex enables the assessment and monitoring of submaximal exercise-induced changes in the cardiovascular system and, consequently, the physical fitness of a given individual.
Table 3: For the categorical predictors, it would be interesting to estimate the differences between categories. ODP: in table 1 and 2, analysing nominal variables, the p-value already express the significance of differents between variable’s categories among boys and girls.
Response: In Tables 1 and 2 which analyzing nominal variables the p-value already denotes the significance of differences between variable’s categories among boys and girls.
Figure 1: I'm a bit confused with this diagram. According to the methods section, you ran linear regressions. Nevertheless, this diagram denotes a SEM model. Linear regressions assume independence of predictors, but here some predictors are clearly related to others. Please, clarify which method is being used to test the relationships of interest in the study.
Response: In order to address the slightly different comments of all reviewers, it was decided to change the main calculation method to a hierarchical regression model. Therefore, the correlation analyses have been completely removed from the Results.
Also, the study has several limitations that should be reckoned by the authors, along with its strengths.
Response: The strengths of our research include the sample size and information on the lifestyle of children and parents. The limitations consist in the lack of measurement of the relationship between parental perception and the actual health and fitness of their children. We plan to address this issue in our future research.
May the authors provide a copy of the questionnaire of the Polish Society of Health, perhaps as supplementary material ar as an Annex? There are lots of variables here for which we do not know the actual definition.
Response: Please see the attached survey of the Polish Society of Health in translation into English
Appendix 1. LIFESTYLE SURVEY
The person filling out the questionnaire:
(a) mother (b) father (c) legal guardian
- How do you assess the overall health of your child?
- Very good b. good c. moderate d. bad e. I find it difficult to estimate
- Does the child suffer from chronic diseases?
÷NO ÷YES, please specify (more than one answer is possible)
- asthma b. hypothyroidism c. enteritis d. allergies (skin, food)
- cardiovascular disease f. respiratory disease g. gastrointestinal disease h. endocrine diseases i. neurological diseases j. other......................................................
- Is the child under the care of a specialist doctor?
Mark which (there may be several answers)
- allergologist b. pulmonologist c. endocrinologist d. cardiologist e. nephrologist f. gastrologist g. neurologist h. other specialist.
- How do you rate the weight of your child?
÷significant underweight ÷ slight underweight ÷ normal ÷ slight overweight ÷ major overweight
- Do you think your child is eating properly?
NO YES I DON'T KNOW
- Do you think the child is active enough during the day?
NO YES I DON'T KNOW
- How often does your child eat sweets and salty snacks?
÷ A few times a day ÷ No more than once a day ÷ A few times a week or less ÷ Not at all ÷ I don't know
- How many times a week does the child eat breakfast (at home)?
÷ Five times a week ÷ 3-5 times a week ÷ 1-3 times a week ÷ The child doesn't eat breakfast ÷ I don't know
- The child usually (at least 3 days a week) goes to school with:
÷ Prepared lunch ÷ Money for lunch ÷ The child doesn't eat lunch at school ÷ I don't know
- How many hours does the child spend every day in front of the television and/or computer? (average for the week) ÷ More than 3 hours ÷ Between 1 and 3 hours ÷ Less than 1 hour
- How many times a week does the child eat fast-food (e.g. hot dogs, hamburgers, fries, kebabs, pizza)? ÷ 5 times a week ÷ 3-5 times a week ÷ 1-3 times a week ÷ She/he doesn't eat fast-food ÷ I don't know
- How many times a week does the child drink sweetened drinks (sodas or flavoured and sweetened water) ? ÷ 5 times a week ÷ 3-5 times a week ÷ 1-3 times a week ÷ I don't know
- How many times a week does the child drink water (bottled water, boiled, tap water)? ÷ 5 times a week ÷ 3-5 times a week ÷ 1-3 times a week ÷ She/he doesn't drink water ÷ I don't know
- How many hours a week does the child exercise or play (excluding physical education classes)?
÷Min. 5 hours a week ÷ Between 3 and 5 hours ÷ Between 1 and 3 hours ÷ Less than 1 hour a week
- Does your child have problems with falling asleep?
÷ No, never ÷ Very rarely ÷ Yes, 1 x a month ÷ Yes, 1-2 x a week ÷ 3-4 times a week ÷ more than 5 times a week
- How often does the child report being tired and sleepy?
÷ Never ÷ Very rarely ÷ Less than 1 time per month ÷ 1-2 times per week ÷ 3-4 times per week ÷ More than 5 times per week
- Are the child's immediate family members overweight or obese? ÷NO ÷YES. if yes, please specify which family members: ÷Mother ÷ Father ÷ Siblings
Have the members of the child's immediate family had any:
- Heart diseases (heart attack, vascular diseases)?
÷NO ÷YES ( ÷Mother ÷ Father ÷ Siblings)
- Strokes?
- Diabetes?
- Lipid disorders (elevated cholesterol)?
- Please give the estimate of mother's weight kg mother's height cm father's weight kg father's height kg pregnancy duration birth weight kg or g
- Mother's education: ÷primary ÷ vocational ÷high school ÷university
- Father's education: ÷primary ÷ vocational ÷high school ÷university
- Has the child been vaccinated according to the vaccination calendar? ÷NOT ÷YES
- Has the child been vaccinated against other diseases (apart from those from the obligatory vaccination calendar, e.g. influenza, pneumoconiosis, pneumoconiosis, etc.)?
NO, YES
Thank you very much for your comments. We appreciate your suggestions concerning methods of critical analysis and evaluation of our research results.
We hope that the above explanation and corrections done have met your expectations
Best regards,
Joanna Ratajczak

Reviewer 2 Report
The introduction is extensive and sufficient. The methods are complete and adequate; however, the results of the tables are incomprehensible. Therefore, it is not possible to validate the information in the text with the tables, nor the discussion. We suggest authors improve the presentation of the results of tables and to resend the manuscript.
The information of tables 1 and 2 is incomplete and, therefore, incomprehensible. Improve the writing titles, present the meaning and units of each variable (means, medians, % ...). The two penultimate columns are unnecessary; this information is already described in methods; in any case, they are presented at the foot of the table. What percentiles are shown, or what do they represent? Both the n and its value must be mentioned for each percentile. The percentages do not carry variances.
The three penultimate columns in table 3 are also not necessary.
In figure 1 of the structural model add the effect of sex.
What do the colors in figure 1 mean?
Author Response
Dear Editor,
Thank you very much for proposed corrections. We have substantially revised the manuscript according to your suggestions. Below are the detailed answers to issues which were raised.
The introduction is extensive and sufficient. The methods are complete and adequate; however, the results of the tables are incomprehensible. Therefore, it is not possible to validate the information in the text with the tables, nor the discussion. We suggest authors improve the presentation of the results of tables and to resend the manuscript
Response: As suggested by the reviewers, the Results section has been improved. Tables 1 and 2 have been changed and a new Table 3 has been created.
The information of tables 1 and 2 is incomplete and, therefore, incomprehensible. Improve the writing titles, present the meaning and units of each variable (means, medians, % ...).
Response: Symbols described presented data was deleted from tables titles and put into tables, next to the variables names, to describe if N and % or M and SD are presented for every variables (N and % for categorical, M and SD for continous). The two penultimate columns are unnecessary; this information is already described in methods; in any case, they are presented at the foot of the table. What percentiles are shown, or what do they represent? Both the n and its value must be mentioned for each percentile. The percentages do not carry variances.
The three penultimate columns in table 3 are also not necessary.
Response: In an effort to address all reviews which contained slightly different comments, two columns containing statistics (X2/Z values and degrees of freedom) have been removed from the Tables, but the last column with p values is still there.
In figure 1 of the structural model add the effect of sex.
Response: In order to satisfy the slightly different comments of all reviewers as fully as possible, it was decided to change the main calculation method to a hierarchical regression model which included the effect of sex. Additional regression was performed to analyze interactions between the gender of the examined children and analyzed predictors of BMI. We believe that this approach will satisfy all Reviewers.
What do the colors in figure 1 mean?
Response: Red color in a Figure 1 denotes the statistical significance of individual predictors, but in the new approach the entire figure has been deleted from the results section.
Thank you very much for your comments. We appreciate your suggestions concerning methods of critical analysis and evaluation of our results.
We hope that our explanations and corrections have met your expectations
Best regards,
Joanna Ratajczak

Reviewer 3 Report
Paper is interesting and well written and only minor changes are needed before it can be accepted for pubblication:
- Introduction is too long and need to be shortened, eventually move some part to the discussion.
- In table 1 age is missing and need to be inserted. On the same table column X2/t and df could be deleted and only p-value mantained.
- Cardiopulmonary capacity is both reported in table 1 and 2. Please correct.
- The variables inside the model should be better described.
- In the introduction also an important recent paper on children and cardiovascular risk factor has been missing and need to be cited, High Blood Press Cardiovasc Prev. 2019 Jun;26(3):191-197.
Author Response
Dear Editor,
Thank you very much for proposed corrections. We have substantially revised the manuscript according to your suggestions. Below are the detailed answers to issues which were raised.
- Introduction is too long and need to be shortened, eventually move some part to the discussion.
Response:.According to the other Reviewers, the Introduction of other reviewers, the introduction is extensive and sufficient, therefore we stick to the original text.
- In table 1 age is missing and need to be inserted. On the same table column X2/t and df could be deleted and only p-value mantained.To ta recenzja, która kaze usuwać 2 kolumny, zostawiajac ze statystyk samo „p”. ODP: information about the age were insert to the table 1 and two columns with unnecessary statistics were deleted from the table.
Response: Information about the age has been put into Table 1 and two columns with unnecessary statistics have been deleted from the table.
- In the introduction also an important recent paper on children and cardiovascular risk factor has been missing and need to be cited, High Blood Press Cardiovasc Prev. 2019 Jun;26(3):191-197.
Response: Thank you very much for this indication. The aforementioned paper, a very valuable contribution to this area of research, is now cited in the manuscript, Introduction, page 2, [20].
Thank you very much for your comments. We appreciate your suggestions concerning methods of critical analysis and evaluation of our results.
We hope that the above explanations and our corrections to the manuscript have met your expectations
Best regards,
Joanna Ratajczak

Reviewer 4 Report
Overall this study provides valuable and thoughtful information about the predictors of obesity among urban girls and boys aged 10 -12 years. However, there are some minor issues in the methods section need to be addressed.
- Generally, different sections of the paper need to be written scientifically. Suboptimal writing skills and number of errors found throughout the paper greatly interfere with clarity and comprehensibility of the paper. Multiple typos, grammatical errors, and some sentences require rephrasing.
- This is a cross-section study. Therefore, the authors have to follow the STROBE statement as some items are still not clarified.
Specific comments:
- Title: Change "obesiti" to "obesity".
- Abstract: line 12, change "p˂0.1" to "p˂0.01".
- Abstract should be more concise without sacrificing important results with p-values. Significant values are very important for the readers.
- The introduction needs to be shortened and many information are repeated. The authors should focus on the significance of the study. Also, the introduction lacked for hypothesis.
- In the methods; regarding the baseline clinical characteristics, please describe how to collect those data with what assessment tools. Who and how to administer the measures to the participants? What were the reliability of anthropometric measurements and physical performance in your current study?
- Methods section needs more details.
- The authors cannot just show the p level, they need to show what type of statistical method used and the actual results of the statistical method which used. The authors missed the key information throughout the results.
Author Response
Dear Editor,
Thank you very much for proposed corrections. We have substantially revised the manuscript according to your suggestions. Below are the detailed answers to issues which were raised
We have introduced the suggested changes in statistical analysis. We also made some linguistic changes with the help of a native speaker.
Specific comments:
- Title: Change "obesiti" to "obesity".
Response: Please see Title
- Abstract: line 12, change "p˂0.1" to "p˂0.01".
Response: Please see: Abstract, line 12
- Abstract should be more concise without sacrificing important results with p-values. Significant values are very important for the readers.
The introduction needs to be shortened and many information are repeated. The authors should focus on the significance of the study. Also, the introduction lacked for hypothesis.
Response: In the opinion of the other Reviewers the Abstract and Introduction are concise enough and that is why we decided not to make any changes to the original version.
- In the methods; regarding the baseline clinical characteristics, please describe how to collect those data with what assessment tools. Who and how to administer the measures to the participants? What were the reliability of anthropometric measurements and physical performance in your current study?
Response: The research included measurements of basic anthropometric indicators, such as: body height (measured with a stadiometer with an accuracy of +/- 0.1 cm), body weight (measured with a doctor's weight with an accuracy of +/- 100 g), physical fitness – using the Kasch Pulse Recovery Test step test, which consists of rhythmic climbing up to a platform 30.5 cm high for 3 minutes. This will allow you to estimate the level of physical fitness on the basis of HR frequency, and thus the level of aerobic physical activity, which is an essential element of a healthy lifestyle. The criteria developed by the International Obesity Task Force (IOTF) were adopted.
The study was conducted by Kułaga, Zb., Różdżyńska, A., Palczewska, I., Grajda A., Gurzkowska B., Napieralska, E, Litwin, M.; and the Research Team OLAF Percentile charts of height, body mass and body mass index in children and adolescents in Poland – results of the OLAF study; Standardy medyczne. Pediatria 2010, 7, 690-700.
Jankowski, M., Niedzielska, A., Brzeziński, M., Drabik, J. Cardiorespiratory Fitness in Children: A Simple Screening Test for Population Studies; Pediatr Cardiol 2015, 36, 27-32.
- Methods section needs more details.
Response: In order to satisfy the slightly different comments of all reviewers as fully as possible, it was decided to change the main calculation method to a hierarchical regression model which included the effect of gender. Additional regression included interactions between the gender of the examined children and the analyzed predictors of BMI. The method section have been enriched accordingly. We believe that this approach to analysis will satisfy all of the reviewers.
- The authors cannot just show the p level, they need to show what type of statistical method used and the actual results of the statistical method which used. The authors missed the key information throughout the results.
Response: Because other Reviewers advised to delete informations about statistics others than p-value, we have removed two columns from Table 1 and 2. Every case, the Pearson’s chi-Square and Mann-Whitney U tests were used to compare boys and girls, which was described in the Methods section. In the first column, next to the name of every analyzed variable, we have provided N (%) for nominal variables compared using the chi-Square test and M (SD) for continuous ones compared with the Mann-Whitney U test.
Thank you very much for your comments. We appreciate your suggestions concerning methods of critical analysis and evaluation of our research results.
We hope that the above explanation and corrections to our manuscript have met your expectations
Best regards,
Joanna Ratajczak

Round 2
Reviewer 1 Report
This is a definitely better version of the manuscript. I am happy with how the authors have taken up my queries and comments and strengthened the manuscript. Still, I have a couple of minor methodological concerns:
- Like I said in my previous report, it is not correct comparing BMI of male and female children using raw values; it's a mistake. You should either use percentiles or z-scores but never raw values. Because you ran models with a dummy variable denoting sex in your main analysis, you don't need standardized BMI for the regressions. However, they are certainly needed for the comparison presented in Table 1. It seems you do have percentiles available, then use percentiles to sex-compare BMI.
- Table 1 should report the participants' mean age. Are they 8-9 years of age or 11-12 years of age? The title is not consistent with the abstract and methods.
- I do not understand how you are dealing with the normality assumption. If your variables are normally distributed, hence you must perform comparisons with the Student's t-test, and report means (SD). Conversely, if they lack a normal distribution, then you must use the Mann–Whitney U test and report median (IQR). What you cannot do is compare variables using a non-parametric test and report mean (SD). Because you have such a huge sample size, you can work under the central limit theorem (CLT), which establishes that, in some situations, when independent random variables are added, their properly normalized sum tends toward a normal distribution. If you do so, you can assume your variables do have a normal distribution, perform parametric analysis (Student's t-test or ANOVA), and report mean (SD). But, please, do not mix apples and oranges.
Author Response
Dear Editor,
Thank you very much for proposed corrections. We have substantially revised the manuscript according to your suggestions. Below are the detailed answers to issues which were raised.
- Like I said in my previous report, it is not correct comparing BMI of male and female children using raw values; it's a mistake. You should either use percentiles or z-scores but never raw values. Because you ran models with a dummy variable denoting sex in your main analysis, you don't need standardized BMI for the regressions. However, they are certainly needed for the comparison presented in Table 1. It seems you do have percentiles available, then use percentiles to sex-compare BMI.
Response: We have changed the BMI calculations from the raw score into percentiles.
- Table 1 should report the participants' mean age. Are they 8-9 years of age or 11-12 years of age? The title is not consistent with the abstract and methods.
Response: We have corrected the error in the title. Age has been added to the Table 1.
- I do not understand how you are dealing with the normality assumption. If your variables are normally distributed, hence you must perform comparisons with the Student's t-test, and report means (SD). Conversely, if they lack a normal distribution, then you must use the Mann–Whitney U test and report median (IQR). What you cannot do is compare variables using a non-parametric test and report mean (SD). Because you have such a huge sample size, you can work under the central limit theorem (CLT), which establishes that, in some situations, when independent random variables are added, their properly normalized sum tends toward a normal distribution. If you do so, you can assume your variables do have a normal distribution, perform parametric analysis (Student's t-test or ANOVA), and report mean (SD). But, please, do not mix apples and oranges.
Response: We have introduced Mdn (IQR).

Reviewer 2 Report
According to the results, the sociodemographic questionnaire is divided into four factors or dimensions, which the authors enter in each step in the hierarchical regression model. This should be mentioned in methods.
Table 1.
The units of height, weight, and BMI are not M (SD) but the following: height in cm, weight in kg, BMI in kg/m2.
They repeat the 95th percentiles twice. Also, with the data they present, the 85-95 percentiles are not below but above.
Similarly, I assume that the height of the parents represents cm, the weight kg, the length of the pregnancy months, and the weight at birth g.
Table 3.
What do the values of the variables represent in each of the 4 models? R2 or adjusted R2?
All of the above must be mentioned in the tables. The tables must be independent and understood without having to go to the text.
Discussions and conclusions
The information on the variance percentages of each model was not corrected in this version. According to the results of the text and table 3, the model with only gender (first factor) predicts less than 1% of the variance, 10.6% that of 7 (second factor), 12.8% that of 11 (third factor), and 14.6% that of 20 variables (four factor). Second paragraph.
I do not understand why the authors eliminate Table 3, since it presents the multiple regression analyzes that they discuss in the second paragraph.
I do not believe that the requested corrections will modify the conclusions, but they will improve the understanding and validity of the manuscript.
Author Response
Dear Editor,
Thank you very much for proposed corrections. We have substantially revised the manuscript according to your suggestions. Below are the detailed answers to issues which were raised.
- According to the results, the sociodemographic questionnaire is divided into four factors or dimensions, which the authors enter in each step in the hierarchical regression model. This should be mentioned in methods.
Response: In line with the Reviewer's suggestion, we have added the information the division of the sociodemographic survey into four dimensions.
- The units of height, weight, and BMI are not M (SD) but the following: height in cm, weight in kg, BMI in kg/m2.
Response:: Units are now entered correctly : height in cm, weight in kg, BMI in kg/m².
- They repeat the 95th percentiles twice. Also, with the data they present, the 85-95 percentiles are not below but above.
Response: We have corrected the last category into “above 95”
4.Similarly, I assume that the height of the parents represents cm, the weight kg, the length of the pregnancy months, and the weight at birth g.
Response: Units are now entered correctly: height of the parents represents cm, the weight kg, the length of the pregnancy months, and the weight at birth g.
5.What do the values of the variables represent in each of the 4 models? R2 or adjusted R2?
Response: Table 3 contains both R2 and adjusted R2, this is mentioned in the table.
- The information on the variance percentages of each model was not corrected in this version. According to the results of the text and table 3, the model with only gender (first factor) predicts less than 1% of the variance, 10.6% that of 7 (second factor), 12.8% that of 11 (third factor), and 14.6% that of 20 variables (four factor). Second paragraph.
Response: In the second paragraph of the discussion we have corrected the text in line with the results.
- I do not understand why the authors eliminate Table 3, since it presents the multiple regression analyzes that they discuss in the second paragraph.
Response: We had to address the remarks of all reviewers. Each of the reviewers had different comments; now gender is a different table.
- I do not believe that the requested corrections will modify the conclusions, but they will improve the understanding and validity of the manuscript.
Response: We agree and we have not changed conclusions.
Thank you very much for your comments. We appreciate your suggestions concerning methods of critical analysis and evaluation of our research results.
We hope that our explanations and corrections have met your expectations.
Best regards,
Joanna Ratajczak
